# *Blautia coccoides* JCM1395^T^ Achieved Intratumoral Growth with Minimal Inflammation: Evidence for Live Bacterial Therapeutic Potential by an Optimized Sample Preparation and Colony PCR Method

**DOI:** 10.3390/pharmaceutics15030989

**Published:** 2023-03-19

**Authors:** Shoko Nomura, Erike W. Sukowati, Yuko Shigeno, Maiko Takahashi, Akari Kato, Yoshimi Benno, Fumiyoshi Yamashita, Hidefumi Mukai

**Affiliations:** 1Laboratory for Molecular Delivery and Imaging Technology, RIKEN Center for Biosystems Dynamics Research, Chuo-ku, Kobe 650-0047, Japan; 2Department of Drug Delivery Research, Graduate School of Pharmaceutical Sciences, Kyoto University, Sakyo-ku, Kyoto 606-8501, Japan; 3Benno Laboratory, RIKEN Baton Zone Program, RIKEN Cluster for Science Technology and Innovation Hab, Wako 351-0198, Japan; 4Department of Pharmaceutical Informatics, Graduate School of Biomedical Science, Nagasaki University, Nagasaki 852-8588, Japan

**Keywords:** bacteria, bacterial cancer therapy, colony PCR, 16S rRNA genes, *Blautia coccoides*

## Abstract

We demonstrate that *Blautia coccoides* JCM1395^T^ has the potential to be used for tumor-targeted live bacterial therapeutics. Prior to studying its in vivo biodistribution, a sample preparation method for reliable quantitative analysis of bacteria in biological tissues was required. Gram-positive bacteria have a thick outer layer of peptidoglycans, which hindered the extraction of 16S rRNA genes for colony PCR. We developed the following method to solve the issue; the method we developed is as follows. The homogenates of the isolated tissue were seeded on agar medium, and bacteria were isolated as colonies. Each colony was heat-treated, crushed with glass beads, and further treated with restriction enzymes to cleave DNAs for colony PCR. With this method, *Blautia coccoides* JCM1395^T^ and *Bacteroides vulgatus* JCM5826^T^ were individually detected from tumors in mice intravenously receiving their mixture. Since this method is very simple and reproducible, and does not involve any genetic modification, it can be applied to exploring a wide range of bacterial species. We especially demonstrate that *Blautia coccoides* JCM1395^T^ efficiently proliferate in tumors when intravenously injected into tumor-bearing mice. Furthermore, these bacteria showed minimal innate immunological responses, i.e., elevated serum tumor necrosis factor α and interleukin-6, similar to *Bifidobacterium* sp., which was previously studied as a therapeutic agent with a small immunostimulating effect.

## 1. Introduction

Live bacterial therapeutics, a treatment approach that uses live bacteria, are recently gaining increasing attention [1]. In addition to using probiotics to improve gut function [2], the development of live bacterial therapeutics against cancer using a tumor-selective growth of injected anaerobic bacteria [3,4] is underway. Clinical trials have been conducted to evaluate the therapeutic efficacy and safety of a few bacterial species and strains, for example, the attenuated *Salmonella typhimurium* VNP20009 [5,6], in cancer therapy. In preclinical settings, another type of live bacterial therapeutics (designer bacteria), which is engineered to produce cytotoxic protein in tumors, is being investigated [7,8,9,10,11]. However, research on live bacterial therapeutics for cancer therapy is still in its infancy, and only a few bacterial species have been examined preclinically, including *Salmonella* [5,6,7,12], *Bifidobacterium* [13,14,15], *Clostridium* [16,17], and *Escherichia* [9,10]. In the past, the application of viral vectors, for example, adenoviruses and retroviruses, in gene therapy caused severe side effects, such as leukemia, which temporarily decreased the momentum for the development of viral-based therapeutics [18,19]. However, new viral vectors, such as lentiviruses and adeno-associated viruses, have been successfully used in gene therapy. Therefore, exploring a wide range of bacterial strains and species that can be used as live bacterial therapeutics is essential for their research and practical application.

To explore promising bacterial species for live bacterial therapeutics, it is necessary to selectively identify and quantify such bacteria. For that, evaluation of antibiotic resistance and labeling of bacteria using reporter proteins, such as fluorescent proteins and luciferases, are frequently used [20,21,22]. However, these approaches can only be used for bacteria with antibiotic resistance and established genetic modification methods. Alternatively, naturally occurring unique DNA barcodes, such as the 16S rRNA gene, can select promising species from a wide range of bacteria. In addition, a colony PCR of the 16S rRNA gene in tissue homogenates is another highly promising, simple, and high-throughput method for screening purposes.

In this study, we first optimized the sample preparation of colony PCR so that even gram-positive bacteria with thick peptidoglycan layers could be lysed without affecting the PCR. Next, we investigated the biodistribution and proliferation of *Blautia coccoides* JCM1395^T^ and *Bacteroides vulgatus* JCM5826^T^, the most dominant, commensal bacteria of the mammalian intestinal flora, in tumors using a colony PCR targeting the 16S rRNA gene of the bacteria after their intravenous administration into mice. Through these investigations, we found that *Blautia coccoides* JCM1395^T^ [23] was highly proliferative in tumors, and we evaluated whether it has any adverse immunostimulatory effects.

## 2. Materials and Methods

### 2.1. Bacterial Cultures

*Bacteroides vulgatus* JCM5826^T^, *Blautia coccoides* JCM1395^T^, and *Bifidobacterium longum* JCM1217^T^ were obtained from the Japan Collection of Microorganisms RIKEN BioResource Research Center (RIKEN BRC, Tsukuba, Japan). The cells were cultured under strict anaerobic conditions without shaking at 37 °C in Gifu anaerobic medium (GAM) broth modified by Nissui (Nissui Pharmaceutical Co., Ltd., Tokyo, Japan), using an AnaeroPack^®^ system (Mitsubishi Gas Chemical Company, Inc., Tokyo, Japan). Competent *E. coli* DH5α cells and pZsGreen were purchased from Takara Bio, Inc. (Kusatsu, Japan). DH5α cells were transformed with pZsGreen by the heat shock method and cultured in a lysogeny broth medium (10 g/L Bacto Tryptone, 5 g/L Bacto yeast extract, 10 g/L NaCl, and 100 μg/mL carbenicillin) with rotary shaking at 130 rpm at 37 °C. 

All bacteria were pre-cultured for 16–18 h until the stationary phase, diluted 20-fold, and cultured for another 2 h for experimental use. The number of CFUs was determined by anaerobic incubation at 37 °C on a GAM agar plate for ~48 h.

### 2.2. Experimental Animals

Male BALB/cCrSlc mice (age: seven weeks) were purchased from Japan SLC Inc. (Hamamatsu, Japan). The murine colon carcinoma cell line (colon-26; RCB2657) was obtained from RIKEN BRC through the National Bio-Resource Project of MEXT, Japan. The cells were cultured in RPMI 1640 medium containing 10% heat-inactivated fetal bovine serum, 100 U/mL penicillin, and 100 μg/mL streptomycin (Nacalai Tesque, Inc., Kyoto, Japan) at 37 °C in a humidified atmosphere containing 5% CO_2_. Colon-26 cells (3 × 10^5^ cells) in 50 μL of HBSS were injected subcutaneously into the right flank to create colon-26 tumor-bearing mice. The mice were used for experiments at around 14 days after tumor implantation, when the tumor volume reached approximately 100 mm^3^. Mice were maintained under the following conditions: 12 h light/12 h dark cycles, continuous feeding, at a room temperature of 22 °C, and 40–60% humidity. All animal experiment protocols were approved by the Ethics Committee on Animal Care and Use of the RIKEN Kobe Institute (authorization number: MA2012-06-17) and conducted in accordance with principles of laboratory animal care (NIH publication No. 85-23, revised 1985).

### 2.3. Bacterial Isolation from Tissues

Bacteria (3.2 × 10^5^–1.6 × 10^8^ CFU) were injected into the colon-26 tumor-bearing mice under 1.5% isoflurane anesthesia. After injection, the mice were euthanized at the indicated times, and the tumor, liver, lung, spleen, and kidney were harvested and weighed. The samples were diluted ninefold with saline, and the tissues were homogenized thoroughly using a Physcotron Handy microHomogenizer (NS-310E; Microtec Co., Ltd., Funabashi, Japan). The tissue homogenates were diluted with saline, seeded onto GAM agar medium, and incubated under strictly anaerobic conditions for 24–48 h.

### 2.4. Colony PCR

Colonies formed on the agar medium were picked with a pipette tip and directly suspended in distilled water in a PCR tube. Glass beads VK01 (M&S Instruments Inc., Osaka, Japan) were added, and the bacteria were lysed by heating at 95 °C for 5 min and vortexing twice for 20 s. The PCR tubes were spun down to remove the glass beads, and 5 units of the restriction enzyme SmaI (Takara Bio Inc., Kusatsu, Japan) was added. After overnight incubation at 37 °C, KOD One PCR Master Mix (TOYOBO Co., Ltd., Osaka, Japan) was added to the resulting samples, and PCR was performed. The primers used are listed in Table 1. The PCR products were run on 2.0% agarose gels. The gels were stained with SYBR^®^ Green I Nucleic Acid Gel Stain (Lonza Co., Ltd., Basel, Switzerland) and imaged using a fluorescence illuminator TCP-20.LM (Berthold Technologies GmbH & Co., Calmbacher, Germany).

### 2.5. Histological Analysis

After the injection of *B. coccoides* JCM1395^T^, the mice were euthanized, and the tumors were collected. The tumors were fixed with 4% paraformaldehyde. Paraffin (3 μm) sections were stained with Gram’s gentian violet solution (Gram staining kit, ab150672) and hematoxylin and eosin (H&E) stain. The stained sections were observed with a built-in 40× objective lens on a Keyence BZ-X710 all-in-one microscope system.

### 2.6. Blood Biochemical Analyses

*B. coccoides* JCM1395^T^ (6.7 × 10^7^ CFU/mouse), *B. longum* JCM1217^T^ (3.0 × 10^7^ CFU/mouse), or *E. coli* (3.4 × 10^7^ CFU/mouse) were intravenously injected into the mice under 1.5% isoflurane anesthesia. The mice were euthanized at the indicated times after injection. Blood samples were collected from the inferior vena cava and allowed to coagulate overnight at 4 °C, and serum was isolated as the supernatant fraction after centrifugation at 10,000× *g* for 10 min. TNF-α and IL-6 levels were determined using the ELISA Ready-set Go! (eBioscience, San Diego, CA, USA), and IFN-γ levels were determined using a Mouse Interferon gamma (IFN-γ) Uncoated ELISA Kit (Thermo Fisher Scientific, Waltham, MA, USA).

### 2.7. Statistical Analyses

GraphPad Prism v. 8 (GraphPad Software, Inc., La Jolla, CA, USA) and IBM SPSS Statistics v. 26 (IBM Corp., Armonk, NY, USA) were used for statistical analyses. Two-group comparisons were analyzed using an unpaired two-tailed *t*-test. One-way or two-way factorial analysis of variance was performed, followed by Dunnett’s or Bonferroni’s multiple comparison test. All *p* values were two-tailed, and a value of *p* < 0.05 was accepted as indicative of statistical significance.

## 3. Results

### 3.1. Optimization of Sample Preparation for Colony PCR of 16S rRNA Gene

The genomic templates for colony PCR were obtained by simple heat treatment of the colony suspension for gram-negative bacteria (*B. vulgatus* JCM5826^T^); however, this method failed for gram-positive bacteria (*B. coccoides* JCM1395^T^ and *B. longum* JCM1217^T^). Adding a surfactant during sample preparation inhibited PCR. In addition, physical lysis using glass beads was performed; the reproducibility of the PCR results were low. However, additional cleavage of genomic DNA by the restriction enzyme *SmaI* resulted in the reproducible detection of PCR amplicons. The final optimized protocol for sample preparation is shown in Figure 1. 

For cultured *B. coccoides* JCM1395^T^ and *B. vulgatus* JCM5826^T^, single bands consistent with the predicted sizes were detected in the amplicons of the colony PCR using the primers for their corresponding 16S rRNA gene (Figure 2A,B). However, we observed ladder bands upon using non-corresponding primers, such as when *B. coccoides* JCM1395^T^ primers were used for the colony PCR of *B. vulgatus* JCM5826^T^ and *B. longum* JCM1217^T^. For each colony on agar medium seeded with a mixture of *B. coccoides* JCM1395^T^ and *B. vulgatus* JCM5826^T^, colony PCR was performed with primers corresponding to the 16S rRNA genes of both species. As a result, the amplicon with one primer showed a single band, and the other showed a ladder band, allowing us to distinguish between them (Appendix A).

### 3.2. Identification of Bacteria in Tumors

Figure 2C,D shows the results of the 16S rRNA gene colony PCR of bacteria colonizing a tumor. The tumor was harvested one week after intravenous administration of a mixture of *B. coccoides* JCM1395^T^ and *B. vulgatus* JCM5826^T^ to a colon-26 tumor-bearing mouse. Of the 5 colonies selected from an agar plate seeded with a 10^5^-fold dilution of the tumor homogenate, 2 showed the characteristic bands of *B. coccoides* JCM1395^T^ described above (Figure 2C, lanes 3 and 4), and 3 showed the bands of *B. vulgatus* JCM5826^T^ (Figure 2D, lanes 1, 2, and 5).

### 3.3. Biodistribution of B. coccoides JCM1395^T^ after Intravenous Administration

To demonstrate the usefulness of this 16S rRNA gene colony PCR for evaluating the biodistribution of gram-positive bacteria, it was applied to evaluate that of *B. coccoides* JCM1395^T^. In a preliminary experiment, bacteria that formed colonies under anaerobic conditions were detected at levels of 10^2^–10^5^ CFU/organ in the organs of colon-26 tumor-bearing mice without the administration of any bacteria (Appendix A). The colony PCR amplicons with the *B. coccoides* JCM1395^T^ 16S rRNA gene primers did not show the characteristic single band of *B. coccoides* JCM1395^T^ (Appendix A–F).

Figure 3A shows the number of CFUs in each organ at 2 h and days 3, 7, and 14 after the intravenous administration of *B. coccoides* JCM1395^T^. In the tumors, the number of CFUs increased to more than 10^7^ CFU/g organ on day 14, while in other organs, it was less than 10^2^–10^4^ CFU/g organ. In the colony PCR of the *B. coccoides* JCM1395^T^ 16S rRNA gene for the tumor homogenates, at 2 h post injection, approximately half of the colonies showed the characteristic single band of *B. coccoides* (Appendix A). All colonies from the tumor samples collected from days 1–14 after the administration of bacteria were identified as *B. coccoides* JCM1395^T^. In contrast, most of the colonies from other organs were not *B. coccoides* JCM1395^T^. H&E and Gram staining images one week after the administration of *B. coccoides* JCM1395^T^ showed central necrosis in the tumors, with a high number of *B. coccoides* JCM1395^T^ at the border between the viable and necrotic areas and a sporadic accumulation of the bacteria within the necrotic area (Figure 3B,C).

### 3.4. Dose Dependency of the Proliferation of B. coccoides JCM1395^T^ in Tumors

Even at a low dose of bacteria (3.2 × 10^5^ CFU/mouse), some mice showed *B. coccoides* JCM1395^T^ proliferation in tumors (Figure 3D). The mean CFU in the tumors increased over 14 days; however, the variation was high. This variability decreased from medium (1.6 × 10^7^ CFU/mouse) to high (1.6 × 10^8^ CFU/mouse) doses of bacterial administration. The *B. coccoides* JCM1395^T^ that reached the tumors 1 h after a high-dose administration were higher in number and proliferated faster than after administration at a low or medium dose, reaching 10^8^–10^9^ CFU/tumor after days 7 and 14. However, the bacterial numbers plateaued around this value with no further growth.

### 3.5. Adverse Effects after Intravenous Administration of B. coccoides JCM1395^T^

The serum levels of tumor necrosis factor-α (TNF-α) and interleukin-6 (IL-6) 1.5 h after *B. coccoides* JCM1395^T^ administration were significantly lower than those after *E. coli* administration (Figure 4A,B). These values were higher than those of the *B. longum* JCM1217^T^ group, which was used as a control having low immunostimulatory effects. However, the difference was not statistically significant. The serum interferon-γ (IFN-γ) level at 6 h was significantly lower than that observed after *E. coli* administration (Figure 4C). It was also lower than that of the *B. longum* JCM1217^T^ group; however, the difference was not statistically significant. 

In addition, none of the low-dose (5.4 × 10^5^ CFU/mouse), middle-dose (2.7 × 10^7^ CFU/mouse), and high-dose (2.7 × 10^8^ CFU/mouse) administered groups showed weight loss (Figure 4D). Although temporary, a 1.5- to 2.0-fold enlargement of the spleen was observed in the middle- and high-dose groups; after 2 weeks of treatment, the spleen weights did not differ significantly from those in the saline-injected control group, except for the high-dose group (Figure 4E).

## 4. Discussion

In this study, we optimized the sample preparation protocol to perform a 16S rRNA gene-targeted colony PCR of both gram-positive and gram-negative bacteria. This method helped us identify bacteria in several organs, including tumors, and evaluated their distribution and proliferation after their administration without any assays to test bacterial genetic modification or drug resistance. Furthermore, we found that *B. coccoides* JCM1395^T^ could be efficiently grown in tumors with minimal adverse effects in mice. 

With our optimized sample preparation for colony PCR, a sufficient amount of genomic DNA was brought into the reaction as a template, regardless of the gram-positive and gram-negative bacteria (Figure 2). Gram-positive bacteria, which do not contain lipopolysaccharide as a membrane component, are potential candidates for live bacterial therapeutics because of their low immunostimulatory effects. Thus, gram-positive bacteria are important targets for evaluation. However, even though the traditional PCR method has been successfully applied on gram-negative bacteria [26,27,28], it is not efficient for gram-positive bacteria. This is because the thick peptidoglycan layers make it difficult to lyse gram-positive bacteria and obtain a sufficient amount of genomic DNA without adding any substances that interfere with PCR. Genomic DNA can be efficiently extracted from *Bifidobacterium* and *Lactobacillus* using 1% Triton-X containing lysis buffers and repeated heating at 100 °C [29]. However, the surfactants can inhibit PCR. Meanwhile, in the colony PCR for the polyhydroxyalkanoate-producing gene of *Bacillus megaterium*, the detection sensitivity was improved by nested PCR, where multiple PCRs were performed on the target sequence [30]. However, this is labor-intensive and time-consuming. In this study, in addition to heat treatment, we physically disrupted bacterial membranes by using glass beads and fragmented genomic DNA by restriction enzymes to improve the PCR efficiency, which helped us perform a reproducible colony PCR even with a gram-positive bacteria, *B. coccoides* JCM1395^T^ (Figure 2). 

A method using 16S rRNA gene amplification by PCR is effective and convenient for the identification of known bacterial species in tissues [25,31]. For example, many periodontal-disease-causing bacteria have been identified by denaturing gradient gel electrophoresis profiles of PCR amplicons using genomes extracted from dental plaque using 16S rRNA gene universal primers, in which several pre-cultured periodontal bacteria were used as comparison specimens [32]. Another technique, the terminal restriction fragment length polymorphism (T-RFLP) analysis, discriminates bacterial species by electrophoresis of restriction-enzyme fragmented 16S rRNA gene amplicons. The accuracy of this method is high since several species of *Helicobacter* spp., such as *H. typhlonicus*, *H. hepaticus*, and *H. bilis*, in mouse feces could be discriminated [33]. In these studies, bacterial genomic DNA was extracted and purified. However, in the present study, as shown in Figure 2A,B, bacterial species were successfully identified by colony PCR, which is a major advancement.

We also detected the presence of commensal anaerobes in the internal organs and tumors of mice at levels of 10^2^–10^5^ CFU/tissue (Appendix A) without any bacterial injection. However, *B. coccoides* JCM1395^T^ was detected in the organs specifically after its administration, and hence, they are not a part of commensal anaerobes in mice. Further, after the administration of *B. coccoides* JCM1395^T^*,* the bacterial abundance increased in tumors by approximately 10^6^-fold (Figure 3A), and all bacterial colonies in the tumors belonged to *B. coccoides* JCM1395^T^ (Appendix A). This result strongly suggests that only the administered *B. coccoides* JCM1395^T^ proliferated in tumors, and it did not stimulate the growth of commensal anaerobes. The selective growth of *B. coccoides* JCM1395^T^ in tumors is in agreement with previous reports on the administration of *Salmonella* [7,12], *Escherichia* [9,10], and *Bifidobacterium* [13,14,15], which are common live bacterial therapeutics for cancer therapy. *B. coccoides* JCM1395^T^ has a proliferative capacity of up to 10^8^ CFU/g tissue in murine tumors. This proliferative capacity is as high or even higher than the proliferative capacity of these common bacteria. In a study, *Bifidobacterium bifidum* grew up to 10^4^–10^5^ CFU/g tissue in Ehrlich ascites tumors after its intravenous injection at 10^6^ CFU/mouse [34], whereas attenuated *Salmonella* strains A1-R and VNP20009 grew up to 10^5^ CFU/g tissue in CT26 tumors after their intravenous injection at 10^5^ CFU/mouse [35] (they could not be injected in higher doses due to side effects). Even SL7207 *Salmonella*, a strain with high proliferative ability in tumors, grew only to 10^8^–10^9^ CFU/g tissue [36,37]. This result was consistent with our study since even *B. coccoides* JCM1395^T^ reached 10^8^–10^9^ CFU/tumor and plateaued thereafter (Figure 3A,D). Moreover, the observation that most of the *B. coccoides* JCM1395^T^ was unevenly distributed at the border between the viable and necrotic areas and in the necrotic area (Figure 3B,C) was also consistent with other studies. For instance, *E. coli* can grow only in limited areas of tumors, probably due to the containment of *E. coli* by neutrophils [38]. The same hypothesis could be true for *B. coccoides* JCM1395^T^, but further study is needed to prove it.

Gram-negative bacteria, such as *Salmonella* and *Escherichia*, carry endotoxins that exert antitumor effects by activating host immunity in tumors [7,21]. However, endotoxins are double-edged swords, as they can induce systemic inflammation beyond tumors. Systematic treatment of CT-26 tumor-bearing mice with 5 × 10^6^ CFU of SL7207 elevated blood TNF-α levels, caused splenomegaly, and resulted in weight loss of approximately 10% [21,37,39]. Even the administration of 5 × 10^6^ CFU of probiotic *E. coli*, such as Nissle 1917, resulted in weight loss in mice for several days after administration [21]. In contrast, the use of *Bifidobacterium* has been reported in anticipation of the low immunogenicity of gram-positive bacteria [14,34,40,41]. Even we observed a significantly lower production of various cytokines after systemic administration of *B. longum* JCM1217^T^ than that of *E. coli* (Figure 4A–C). Bacteria such as *B. longum* JCM1217^T^ are not likely to exert antitumor effects on their own due to their mild immunostimulatory effects; however, some functionalization approaches can impart antitumor effects to them. For example, genetically modified strains expressing drug-converting enzymes, such as cytosine deaminase [14] and cytotoxic proteins [42], have been developed. However, *Bifidobacterium* requires repeated administration of lactulose to promote its growth in tumors [34], which is one of the obstacles to its clinical application. In contrast, *B. coccoides* JCM1395^T^ showed a high tumor-selective proliferative ability without requiring additional nutrition after administration, even though its immunogenicity was as low as that of *Bifidobacterium* (Figure 4A–E). At present, no simple genetic modification technology for *B. coccoides* JCM1395^T^ has been reported, and additional research is needed for therapeutic functionalization. Overall, *B. coccoides* JCM1395^T^ is a promising candidate for live bacterial therapeutics for cancer treatment.

The colony PCR targeting 16S rRNA gene could effectively evaluate the ratio of each bacterium in tumors after multiple bacterial species were administered (Figure 2C,D). Therefore, this method can be extended to a wide selection of bacterial species that efficiently grow in tumors from bacterial libraries. In this method, primers must be able to be designed; it is difficult to apply it to bacteria for which 16S rRNA gene sequence information is not available, though. For the clinical application of a bacteria in cancer therapy, the bacteria may need to be administered multiple times to the same patient. In such cases, the patient may acquire immunity to the bacterial species, leading to severe adverse effects or reduced therapeutic efficacy. Furthermore, the bacteria to be used in live bacterial therapeutics for cancer therapy should be efficient in not only intratumoral growth, but also immunostimulatory ability and the production of anti-cancer factors [43]. From this perspective, future research should be conducted to increase the range of bacteria for live bacterial therapy in cancer. 

In conclusion, we developed a protocol for 16S rRNA gene-targeted colony PCR for both gram-positive and gram-negative bacteria. As genetic modification is not required, this method can be applied to evaluate the intra-tissue proliferative ability of a wide range of bacteria. This will increase the library sizes of the potential bacteria for live bacterial therapeutics. We believe that exploring bacterial species with high proliferative ability in tumors and minimal inflammation, such as *B. coccoides* JCM1395^T^, will advance the applicability of live bacterial therapeutics in clinical settings.

## Figures and Tables

**Figure 1 pharmaceutics-15-00989-f001:**
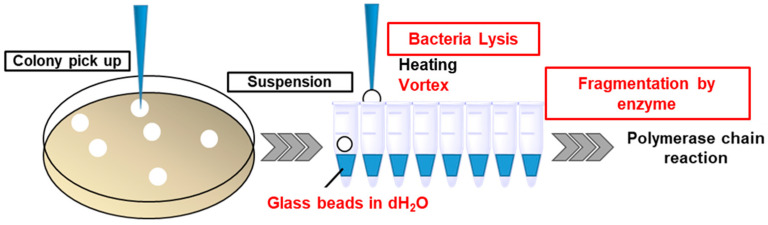
Optimized preparation protocol of DNA templates for colony PCR. Red highlighting indicates the optimized steps.

**Figure 2 pharmaceutics-15-00989-f002:**
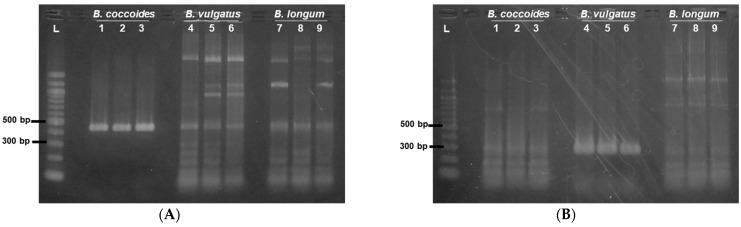
Identification of *B. coccoides* JCM1395^T^ and *B. vulgatus* JCM5826^T^ by colony PCR. SYBR^®^ GREEN stained agarose gel electrophoresis images applied with 16S rRNA gene PCR amplicons by primers corresponding to (**A**) *B. coccoides* JCM1395^T^ and (**B**) *B. vulgatus* JCM5826^T^. The PCR amplicons from *B. coccoides* JCM1395^T^*, B. vulgatus* JCM5826^T^*,* and *B. longum* JCM1217^T^ colonies were applied to lanes 1–3, lanes 4–6, and lanes 7–9, respectively. (**C**,**D**) Detection of *B. coccoides* JCM1395^T^ and *B. vulgatus* JCM5826^T^ from tumor homogenates by colony PCR. The experiments were performed on three mice, respectively, of which typical gel images are shown. SYBR^®^ GREEN stained agarose gel electrophoresis images applied with 16S rRNA gene PCR amplicons of 5 colonies picked from the agar plate seeded with tumor homogenate by primers corresponding to (**C**) *B. coccoides* JCM1395^T^ and (**D**) *B. vulgatus* JCM5826^T^. Tumors were isolated from colon-26 tumor-bearing mice a week after receiving the bacterial mixture of *B. coccoides* JCM1395^T^ (3.7 × 10^7^ CFU/mouse) and *B. vulgatus* JCM5826^T^ (5.4 × 10^7^ CFU/mouse). The number in the figure indicates each colony.

**Figure 3 pharmaceutics-15-00989-f003:**
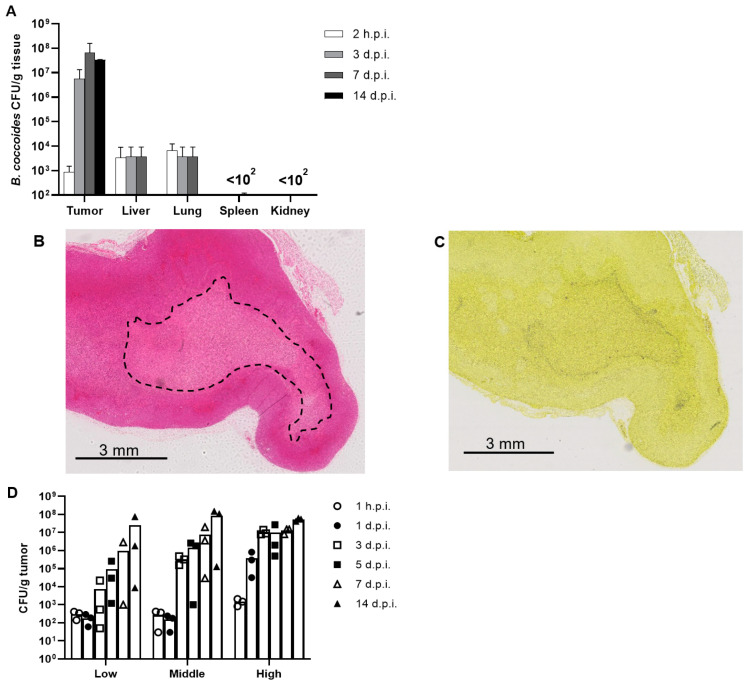
Biodistribution of *B. coccoides* JCM1395^T^ after its intravenous administration. (**A**) Biodistribution of intravenously administered *B. coccoides* JCM1395^T^ (6.8 × 10^7^ CFU) in colon-26 tumor-bearing mice; CFU values were corrected based on the numbers of colonies showing a single band of *B. coccoides* JCM1395^T^ by colony PCR. Each value represents the mean ± SD (*n* = 3–4). Tumor sections with (**B**) H&E staining and (**C**) Gram staining at seven days after *B. coccoides* JCM1395^T^ injection (2.7 × 10^7^ CFU). Original magnifications were at ×1. The dotted line indicates necrotic area. *B. coccoides* JCM1395^T^ was tinted purple. (**D**) Dose dependency of *B. coccoides* JCM1395^T^ proliferation in tumors. *B. coccoides* JCM1395^T^ numbers in CFU/tumor at 1 h and days 1, 3, 5, 7, and 14 from intravenous injection of low-dose (3.2 × 10^5^ CFU), medium-dose (1.6 × 10^7^ CFU), and high-dose (1.6 × 10^8^ CFU) *B. coccoides* JCM1395^T^ are shown. Each point represents the value from one mouse and the bars represent the mean for each group (*n* = 3).

**Figure 4 pharmaceutics-15-00989-f004:**
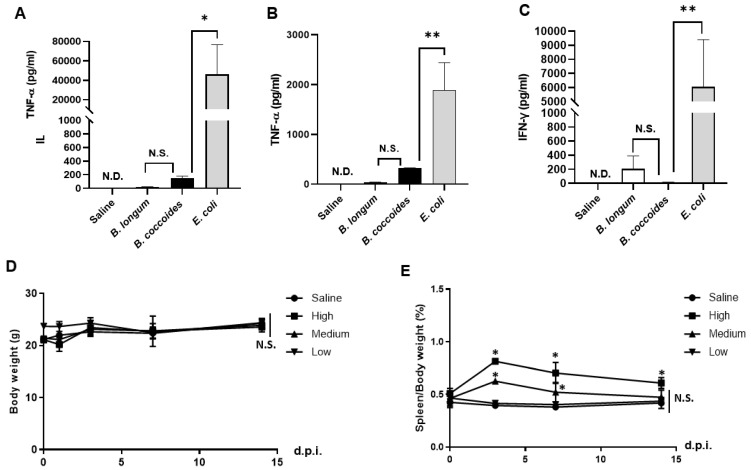
Adverse effects after intravenous administration of *B. coccoides* JCM1395^T^. (**A**) Tumor necrosis factor-α and (**B**) interleukin-6 concentration in serum at 1.5 h, and (**C**) interferon-γ concentration in serum at 6.0 h after intravenous injection of saline, *B. coccoides* JCM1395^T^ (6.7 × 10^7^ CFU), *B. longum* JCM1217^T^ (3.0 × 10^7^ CFU), and *E. coli* (3.4 × 10^7^ CFU). (**D**) Body weight and (**E**) spleen weight changes after intravenous injection of low-dose (5.4 × 10^5^ CFU), medium-dose (2.7 × 10^7^ CFU), and high-dose (2.7 × 10^8^ CFU) *B. coccoides* JCM1395^T^. * *p* < 0.05; ** *p* < 0.01. Each value represents the mean ± SD (*n* = 3). N.D., not detected. N.S., not significant.

**Table 1 pharmaceutics-15-00989-t001:** List of 16S rRNA gene-targeted primers used in this study.

Species	Oligonucleotide Sequence (5′→3′)	Amplicon Size	Reference
*Blautia* *coccoides* JCM1395^T^	F: 5′-AAATGACGGTACCTGACTAA-3′ R: 5′-CTTTGAGTTTCATTCTTGCGAA-3′	438 bp	Kurakawa et al. (2015) [24]
*Bacteroides* *vulgatus* JCM5826^T^	F: 5′-GCATCATGAGTCCGCATGTTC-3′ R: 5′-TCCATACCCGACTTTATTCCTT-3′	267 bp	Wang et al. (1996) [25]

## Data Availability

The data used to support the findings of this study are available from the corresponding author upon reasonable request.

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
