# Peer review of "Blautia coccoides JCM1395T Achieved Intratumoral Growth with Minimal Inflammation: Evidence for Live Bacterial Therapeutic Potential by an Optimized Sample Preparation and Colony PCR Method"

_pharmaceutics, 2023, doi:10.3390/pharmaceutics15030989_

Round 1

Reviewer 1 Report

In this study, the authors optimized the method of sample preparation for colony PCR for bacteria Gram+  and studied biodistribution and bacterial proliferation in tumors through PCR 16S rna gene of bacteria after their intravenous administration in mice.

The paper is interesting but some improvements must be made...

1)Authors should modify the abstract to make it more attractive, emphasizing more the results obtained

2)Why did you select only male mice?

3)Authors should clarify the conditions of maintenance of mice such as feeding, dark light cycle...

4)In some graphs of the results lacks statistics. Apply it, please

5)Authors should better discuss the results in the discussion by referring to the reading, where it is possible.

Reviewer 2 Report

In the article entitled “Blautia coccoides JCM1395T achieved intratumoral growth with minimal inflammation: evidence for live bacterial therapeutic potential by an optimized sample preparation and colony PCR 4 method.  However, it needs modifications and few explanations.

1. The abstract is not well written. Avoid description of methods and try to put few interesting results to attract the readers.

2. The introduction should be more compact with few explanations: a. why these bacteria have been chosen? b. The relation between these bacteria and tumors. c. The relevance and novelty of this study.

3. These are very preliminary observations and needs more experiments specially a transcriptomics experiment to understand the global gene regulation to support the therapeutic potential.

4. Colony PCR is a very common method in bacteriology. So, I am not able to understand the importance of this section? Please explain.

5. In mice experiment please mention the number (n=?).

6. Authors have used three bacteria, but there are only two primers. ????

7. Figure 2 is not clear enough and the resolution must be enhanced. In A and B, there are multiple bands. It needs justifications.

8. Please correct the figure 3A axis. CFU/tissue does not mean anything. The calculation should be CFU/g tissues.

9. The discussion section is not compact and should be improved.

10. Authors have measured only three serum parameters which is not sufficient to conclude the statement. At least 10 blood parameters should be measured.

11. Please check the references carefully. Scientific names must be in italics.  

Reviewer 3 Report

The paper by Shoko Nomura and co-workers reports the development of a protocol for 16S rRNA gene-targeted colony PCR for tumor-targeted bacteria screening. Further, they demonstrate that Blautia coccoides JCM1395T has the ability of proliferating in tumors by use of this method. Considering that identification of promising bacterial species for live bacterial therapeutics is necessary, the method in this paper is helpful. However, some minor issues about the method in this paper need to be addressed. As an overall impression, publication of this work can be considered after addressing some minor concerns.

 1, Page 4, Figure 1, the optimized preparation protocol of DNA templates for colony PCR in this paper are described in this figure. It is better to add the protocol of colony PCR before optimization. The comparison of two methods in one figure would be more readable.

2, This method that using 16S rRNA gene amplification by PCR seems to suit for the identification of known bacterial species in tumor.  Could this method have potential to be developed for identification the unknown bacterial species in tumor? Or how to screen the unknown bacterial species that have potential to proliferate in tumor? Please discussed in Discussion section.

3, Page 6, Figure 3, the figure of H&E staining and Gram staining are not clear. High resolution pictures should be provided. It is better to draw arrows to point out the bacteria, the viable area and necrotic area.

4, Page 7, Line 236, the sentence of line 236-240 is break.
